# Non-Invasive Cardiac Output Determination Using Magnetic Resonance Imaging and Thermodilution in Pulmonary Hypertension

**DOI:** 10.3390/jcm11102717

**Published:** 2022-05-11

**Authors:** Lindsey A. Crowe, Léon Genecand, Anne-Lise Hachulla, Stéphane Noble, Maurice Beghetti, Jean-Paul Vallée, Frédéric Lador

**Affiliations:** 1Division of Radiology, Diagnostic Department, Geneva University Hospitals, 1205 Geneva, Switzerland; lindsey.crowe@hcuge.ch (L.A.C.); anne-lise.hachulla@hcuge.ch (A.-L.H.); jean-paul.vallee@hcuge.ch (J.-P.V.); 2Faculty of Medicine, University of Geneva, 1205 Geneva, Switzerland; stephane.noble@hcuge.ch (S.N.); maurice.beghetti@hcuge.ch (M.B.); frederic.lador@hcuge.ch (F.L.); 3Pulmonary Hypertension Program, Geneva University Hospitals, 1205 Geneva, Switzerland; 4Division of Pulmonary Medicine, Department of Medicine, Geneva University Hospitals, 1205 Geneva, Switzerland; 5Division of Cardiology, Department of Medicine, Geneva University Hospitals, 1205 Geneva, Switzerland; 6Paediatric Cardiology Unit, Geneva University Hospitals, 1205 Geneva, Switzerland; 7Centre Universitaire Romand de Cardiologie et Chirurgie Cardiaque Pédiatrique, University of Geneva and Lausanne, 1205 Geneva, Switzerland

**Keywords:** pulmonary hypertension, cardiac output, magnetic resonance imaging, thermodilution

## Abstract

Magnetic resonance imaging (MRI) can be used to measure cardiac output (CO) non-invasively, which is a paramount parameter in pulmonary hypertension (PH) patients. We retrospectively compared stroke volume (SV) obtained with MRI (SV_MRI_) in six localisations against SV measured with thermodilution (TD) (SV_TD_) and against each other in 24 patients evaluated in our PH centre using Bland and Altman (BA) agreement analyses, linear correlation, and intraclass correlation (ICC). None of the six tested localisations for SV_MRI_ reached the predetermined criteria for interchangeability with SV_TD_, with two standard deviations (2SD) of bias between 24.1 mL/beat and 31.1 mL/beat. The SV_MRI_ methods yielded better agreement when compared against each other than the comparison between SV_MRI_ and SV_TD_, with the best 2SD of bias being 13.8 mL/beat. The inter-observer and intra-observer ICCs for CO_MRI_ were excellent (inter-observer ICC between 0.889 and 0.983 and intra-observer ICC between 0.991 and 0.999). We could not confirm the interchangeability of SV_MRI_ with SV_TD_ based on the predetermined interchangeability criteria. The lack of agreement between MRI and TD might be explained because TD is less precise than previously thought. We evaluated a new method to estimate CO through the pulmonary circulation (COp) in PH patients that may be more precise than the previously tested methods.

## 1. Introduction

Pulmonary hypertension (PH) is defined as a resting mean pulmonary artery pressure (mPAP) ≥ 25 mmHg measured by right heart catheterisation (RHC) [1]. Five groups with different aetiologies can be differentiated. Pulmonary arterial hypertension (PAH) is a rare pulmonary vasculopathy diagnosed after exclusion of other class of PH (groups 2, 3, 4, and 5) and confirmed with RHC showing a precapillary PH (PcPH) that is characterised by an mPAP ≥ 25 mmHg and a pulmonary artery wedge pressure (PAWP) ≤ 15 mmHg [1]. Group 4 is defined as an obstruction of the pulmonary arteries leading to PH. The main entity of this group is called chronic thrombo-embolic pulmonary hypertension (CTEPH).

Cardiac output (CO) measurement is important for the prognosis of patients with PAH via risk class stratification. It is also essential for diagnoses using the calculation of pulmonary vascular resistance (PVR). A proposal suggested redefining PH with a lower mPAP threshold of >20 mmHg and the inclusion of PVR > 3 WU as an obligatory criterion for PcPH [2]. In this context, a reliable non-invasive method to measure CO could be valuable for the patients and the clinician in order to avoid some invasive procedures.

Invasive methods are acknowledged to be the methods of choice to measure CO [1]. Direct Fick (DF) is the historical gold standard [1]. However, CO measurement using DF is cumbersome due to the measurement of oxygen consumption. Thermodilution (TD) has been accepted as the reference method in the current international guidelines [1]. This proposal was based on one study showing a good agreement between TD and DF [3]. TD is nowadays the most used method for the determination of CO for PH patients. Indirect Fick (IF) lacks reliability, and poor agreement with DF or TD was shown both retrospectively and prospectively in large mixed PH population [4,5,6].

Many different non-invasive methods to measure CO have been studied in precapillary PH patients including bioimpedance, bioreactance, inert gas rebreathing, and pulse wave analysis [3,6,7,8,9,10,11,12]. None of the studied non-invasive method reached predetermined criteria for interchangeability with an invasive method (DF or TD) in a recent systematic review [13]. CO determined with transthoracic echocardiography (TTE) also showed poor performance in comparison to TD in a meta-analysis [14].

MRI is acknowledged as an important emerging tool in PH management. It allows anatomical evaluation of the right ventricle (RV) and the pulmonary artery (PA), offers some new prognostic variables, and can be helpful for the diagnosis and classification of PH in specific cases [15,16,17,18]. Its role in the determination of CO remains under investigation. SV determination using flow measurement in the ascending aorta (AAO) (SV_AAO_) and volumetric assessment of the left ventricle (LV) (SV_LV_) agreed closely with the SV derived from the DF (SV_DF_) in 32 PAH patients with a 2SD of bias of +/−7.5 mL/beat and +/−9.6 mL/beat, respectively, with no significant bias [19]. However SV derived with flow measurement in the pulmonary artery (SV_PA_) and volume assessment of the right ventricle (SV_RV_) showed poor agreement with SV_DF_ [19]. To the best of our knowledge, no methods have shown the ability to precisely calculate CO through the pulmonary circulation (COp) in precapillary PH.

Our aims were to determine (1) the agreement between SV_MRI_ and SV_TD_ in a population evaluated in our PH centre; (2) the inter- and intra-observer reproducibility, and the agreement between the different SV_MRI_ methods in the same population. We hypothesised that a new strategy for COp estimation based on the sum of the right pulmonary artery (RPA) and the left pulmonary artery (LPA) SV (SV _(RPA+LPA)_) would be more precise than the localisations tested thus far in PH patients (SV_PA_ and SV_RV_).

## 2. Materials and Methods

### 2.1. Study Population and Setting

We retrospectively included patients in our PH centre evaluated with both RHC and MRI imaging. We included all patients from variable settings, including (1) diagnosis evaluation of suspected PH, (2) follow-up of PH, and (3) follow-up of PH patients after treatment. Patients were excluded if (1) the delay between RHC and MRI was more than 3 weeks, (2) their age was <18 years, (3) a clinical deterioration occurred between RHC and MRI, and (4) in the case of concomitant pregnancy. The local ethical committee approved our study (2017-00716).

### 2.2. Right Heart Catheterisation

The RHC exam was performed in a supine position with continuous monitoring of the electrocardiogram and arterial oxygen saturation using pulse oximetry (SpO_2_). The mean systemic arterial pressure was measured at the brachial artery with an automatic inflating cuff. The modified Seldinger technique was used for venous catheterisation of the femoral, basilic, or cephalic vein with a 7F Terumo Glidesheath Slender radial introducer sheath (Terumo, Tokio, Japan), which has an outer diameter of a 6F sheath. The Swan Ganz catheter (7F) allowed the resting haemodynamic evaluation and included mPAP, PAWP, and CO_TD_. The mid-thoracic line was used for the zero-level reference.

PVR, cardiac index (CI), and stroke volume (SV) were calculated with the respective formulae: PVR = (mPAP − PAWP)/CO_TD_; CI = CO/(body surface area); and SV = CO/(heart rate). 

#### CO Determined by TD

TD was performed with 10 mL of iced, cold, sterile isotonic glucosaline solution injected in the proximal catheter’s lumen. The temperature change was recorded at the distal end of the probe with a thermistor. Measurements were performed in triplicate, and the mean value was recorded if the difference between the highest and lowest value was ≤10%. Otherwise, two more measurements were performed with the deletion of the highest and lowest values. The mean of three remaining values was then calculated.

### 2.3. CO Determined by MRI

The MRI was acquired on a clinical Siemens 1.5 T AERA and 3 T PRISMA FIT (Siemens, Erlangen, Germany) and analysed using SyngoVia software from Siemens (Siemens, Erlangen, Germany). The CO_MRI_ measurements were made with flow velocity analysis in (1) AAO, (2) PA, (3) RPA + LPA, and (4) descending aorta (DAO) + superior vena cava (SVC), and the volumetric measurements were performed during the systole and diastole of (5) RV and (6) LV. The SV was then derived from the CO measurement (SV = CO/heart rate). The averaged heart rate was measured from the electrocardiogram (Siemens, Erlangen, Germany) used for the synchronisation of the MRI acquisitions. In the absence of a shunt or valvular leak, all of the 6 SV_MRI_ estimations should reflect the same value. The images were acquired for the flow analysis using phase-contrast 2D FLASH with the image plane perpendicular to the flow direction, a slice thickness of 6 mm, and a 1.5–2 mm in-plane resolution according to the patient size, with a typical velocity encoding 250 cm/s for the aorta, 150 cm/s for the pulmonary arteries, and 120 cm/s for the superior vena cava (which was increased in the case of aliasing). The TR/TE was 28–37/2.5 ms; the flip angle was 20; the bandwidth was 450 Hz; and there were 2–3 signal averages according to the breathing pattern. For the volumetric analysis, a 2D true-FISP cine sequence was used with the following typical parameters: retrospective ECG gating, a resolution of 1.5 mm × 1.5 mm, a slice thickness of 8 mm, GRAPPA acceleration factor 3, TE of 1.3 ms, TR of 24 ms, flip angle of 30, with the left and right 2-chamber, 4-chamber, and contiguous short axes covering the whole heart.

Two different experienced investigators estimated CO_MRI_. They were unaware of the CO_TD_ measurements when analysing the MRI data. 

### 2.4. Statistics

The statistical analysis was conducted with SPSS (version 21, IBM, Armonk, New York). The data are given in mean +/− standard deviation (SD) unless otherwise stated. Statistical significance was defined as *p* < 0.05. Linear correlations were determined using linear regression with slope and intercept (y = ax + b), and the coefficient of correlation (r) was calculated. Bland and Altman (BA) analysis was used to determine bias (mean difference of SV), limits of agreement (LoA), and percentage error (PE) when comparing SV_MRI_ against SV_TD_ and the 6 SV_MRI_ against each other. The LoA were calculated as bias +/−2SD of bias. PE for SV was calculated as (2SD of bias/mean SV) × 100. We used the predetermined 2SD of bias of 17.9 mL/beat and a PE of 30% to accept interchangeability between the two methods of SV determination [20]. The cut-off of 17.9 mL/beat was calculated from the proposed predetermined CO 2SD of bias of 1.25 L/min (1250 mL/min) and a mean heart rate in the cohort of 70 beats/min [20].

The 6 estimations of SV_MRI_ were compared to SV_TD_ and against each other. The SV_MRI_ agreement between the methods comprised 15 pairs of measurements (the 6 methods compared against one another). SV was used for the comparison between different methods because SV is more stable than CO when two methods are not performed exactly at the same time due to possible variation of the HR [19]. Indeed, while HR variations will influence CO, SV is more stable in PAH even with exercise [21]. 

The results of the heart rate of the different tests (RHC and MRI) were compared using Student’s *t* test.

Inter- and intra-observer reproducibility was assessed using intraclass correlation (ICC) for the MRI CO measurements using flow assessment (AAO, DAO + SVC, RPA + LPA, and PA) in 10 randomly selected patients. For inter-observer ICC, the measurements were made independently by two different investigators (L.C. and A-L.H.). For intra-observer ICC, the same investigator (L.C.) made the measurements separated by a period of two months.

## 3. Results

### 3.1. Patients 

A total of 41 patients met the inclusion criteria. A total of 17 patients were excluded, mainly due to excessive delays between RHC and MRI (*n* = 12). The flow chart is given in Figure 1. A total of 24 patients were included for the analysis, including 9 males and 15 females aged 60 ± 14 years at time of MRI (range 20–79 years). A total of 12 patients had PH during RHC (group 1, *n* = 4; group 4 (CTEPH), *n* = 7; group 2, *n* = 1). A total of 12 patients had no PH during RHC (no PH, *n* = 3; treated CTEPH, *n* = 5, including four patients after pulmonary endarterectomy and one patient after balloon pulmonary angioplasty; chronic thrombo-embolic pulmonary disease (CTEPD), *n* = 2; and treated group 1 PAH, *n* = 2). The time between RHC and MRI was 6 ± 6 days (range 0–21). The patients’ details are given in Table 1.

Flow chart of the patients. Patients who were followed up in our PH centre who benefited from both right heart catheterisation and cardiac magnetic resonance were screened for inclusion. After exclusion of patients not fulfilling the predetermined criteria, we included 24 patients in the final analysis. MRI = magnetic resonance imaging. 

### 3.2. CO Measurement Using MRI

Figure 2 shows the placement of images for the major vessels and the resulting phase-contrast flow image, followed by the calculated flow curve. 

Inter- and intra-observer reproducibility (single-measure ICC) for all MRI flow quantification ranged from 0.826 to 0.983 and 0.866 to 0.999, respectively, as shown in Table 2. 

### 3.3. Comparison between MRI and TD

The mean heart rate did not differ between the RHC and MRI as confirmed by the paired *t*-test with *p* = 0.4.

Table 3 summarizes the mean and SD values for all the parameters. Overall, the mean SV was 70 mL (SD = 20). 

Figure 3 shows the BA analyses of the six SV_MRI_ methods against SV_TD_. Table 4 summarises the BA analyses of the six SV_MRI_ against SV_TD_ as well as the coefficient of correlation and linear regression. The 2SD of bias ranged from 24.1 to 31.1 mL/beat, and the bias ranged from −2.9 to −11.3 mL/beat. PE ranged from 34.9% to 42.8%.

### 3.4. Comparison between Different MRI Methods

Figure 4 shows the Bland and Altman analysis of SV_(RPA+LPA)_ compared with (1) SV_AAO_ and (2) SV_(DAO+SVC)_ showing narrow LoA and a small bias. SV_(RPA+LPA)_ compared with SV_(DAO+SVC)_ yielded the best agreement, with a bias of 4.1 mL/beat, a 2SD of bias of 13.8 mL/beat, and a PE of 19.7%. The agreement between SV_(RPA+LPA)_ compared with SV_AAO_ was also very good, with a bias of −2.1 mL/beat, a 2SD of bias of 17.9 mL/beat, and a PE of 25.5%. Table 5 summarises the Bland and Altman analysis of the comparisons between the six different MRI methods, as well as the coefficient of correlation and linear regression. The bias ranged from −6.8 to +8.6 mL/beat; the 2SD of bias ranged from 13.8 to 29.1 mL/beat; and the PE ranged from 19.7% to 39.4%. 

## 4. Discussion

In this study, we showed that (1) SV_MRI_ was not interchangeable with SV_TD_ using predetermined criteria; (2) SV_(RPA+LPA)_ yielded the best agreement with the other SV_MRI_ methods including the already validated SV_AAO_ method; and (3) the agreement between the different SV_MRI_ methods was globally better than when SV_MRI_ was compared against SV_TD_.

### 4.1. Regarding the Statistical Analysis

BA graphs are the analysis of choice when two methods measuring the same variable are compared. They provide information on the degree of agreement between the compared methods [22]. For proper BA analysis, a specific cut-off for the acceptance of the interchangeability of two methods is mandatory, otherwise this choice is left to the subjectivity of the authors [22]. In the field of anaesthesiology and intensive care medicine, a 2SD of bias of 1 L/min or a PE of 20% when comparing a new CO estimation method to the gold standard (DF), or an LoA of 1.25 L/min or PE of 30% when comparing a new method to a reference method (TD) that is not the gold standard, have been suggested [20,23]. The use of a wider LoA and PE when comparing a new method to a reference method (TD) are suggested because the LoA and PE are the results of the intrinsic imprecision of both methods with a reference method that is supposed to be less precise than the gold standard [20]. By precision, we refer to how close the values of repeated measurements are [20]. The predetermined cut-off should be determined based on the clinical purpose of CO measurement, which will influence the degree of precision needed. In the field of PH, a cut-off has never been proposed [13]. In this context, we admitted it was reasonable to rely on an existing cut-off (1.25 L/min, with a derived cut-off for SV of 17.9 mL/beat) even though it might not be the ideal cut-off for the clinical context studied. 

### 4.2. Regarding Interchangeability between MRI and TD

None of the six SV_MRI_ reached the predetermined interchangeability criteria when compared against SV_TD_. Therefore, in our population, SV_MRI_ cannot be considered interchangeable with the reference SV_TD_ method. This is concordant with previous results in the field showing a wide range of the 2SD of bias (ranging from 1.9 L/min to 2.4 L/min) and PE when comparing MRI with TD [24,25,26,27]. As aforementioned, the LoA and PE are the results of the imprecision of the two studied methods. The LoA and PE were globally smaller when compared between different MRI measurement methods than when we compared MRI with TD. This raises the question of whether the actual reference method (TD) could be the cause of the observed lack of agreement. TD is a reference method based on an analysis of 35 PcPH patients in 1999, which showed good agreement between TD and DF with a 2SD of bias of 1.1 L/min [3]. However, recent published data have shown that the agreement between DF and TD is probably lower than previously thought, with a wide range of PE (42% and 44.6%) and LoA (2SD of bias of 2.48 L/min), leading to misclassification in the prognosis assessment of patients with PAH and misclassification of exercise PH [28,29,30]. In this context, it is likely that TD might be the cause of the lack of agreement when we compared TD with MRI. This would also explain the constant lack of agreement between CO_MRI_ and CO_TD_ in other studies in this field, while a good agreement between SV_AAO_ and SV_RV_ was demonstrated against the gold standard SV_DF_ [19,24,25,26,27]. In our study, the delay between MRI and TD could also contribute to lower the observed agreement between MRI and TD due to change in the haemodynamic condition of the patients. The exclusion of patients with conditions that could lead to rapid change in haemodynamic (e.g., pregnancy and clinical deterioration between RHC and MRI) probably lowered this potential effect. Since we obtained similar results to those of previous studies in the field, we do not think that the delay made a significant change in the observed agreement.

### 4.3. Regarding the Comparison between Different MRI Methods 

We provided an assessment of six different types of SV_MRI_ data, including measurements of flow in localisations such as (LPA + RPA) and (SVC + DOA) that have never been tested in this population thus far. COs and COp were both measured in three localisations i.e., (AAO, DAO + SVC, and LV) and (RV, PA, and RPA + LPA). All of these CO estimates and SV derivates should be equal in the absence of a significant shunt. 

One issue related to MRI in the evaluation of PH is the lack of a method to evaluate the COp. Indeed, SV_RV_ and SV_PA_ lacked agreement with the gold standard DF [19]. SV_RV_ could be less precise in PH patients due to (1) tricuspid insufficiency, (2) large trabeculations in the RV, and (3) the complicated anatomy of the RV with difficult delimitation of the inner border of the cavity. SV_PA_ imprecision could be due to (1) the presence of a vortex and non-laminar flow in dilated PA with an irregular border and (2) pulmonary regurgitation.

COp could be a valuable measurement, especially for PH populations, because it could allow the measurement of shunts between the pulmonary and systemic circulation. Congenital heart disease, including shunts, are one of the most common causes of PAH in the adult population [1]. The measurement of COp in patients with shunts could also allow the calculation of the correct PVR, which is necessary for diagnosis and treatment decisions in some of these patients, including discussion about the shunt’s closure. For PVR calculation, mPAP and PAWP measurements are necessary. MRI can provide an estimation of mPAP and PAWP, even though the methods used require further validation [31]. Furthermore, a new method to determine COp is greatly needed because the TD and Fick methods are known to lack precision, especially in the case of shunts and extreme CO [32]. 

In our study, we found that SV_PA_ yielded poor results, which is consistent with previous publications and is probably caused by the reasons explained above [19]. SV_RV_ yielded results similar to those of other methods of SV determination and was not worse than SV_AAO_ or SV_LV_ for SV estimation, as had been previously shown [19]. This could be explained by the mildly elevated mPAP in our cohort that is associated with a low degree of tricuspid regurgitation. Anatomical determination of the RV’s contour is also becoming easier with the improvements in MRI techniques and analysis software. Tricuspid regurgitation might be the main mechanism leading the lack of agreement of the method. Indeed, the volume moving forward through the pulmonary valve or returning back through the tricuspid valve cannot be differentiated. This is of interest as SV_RV_ could be a method used to calculate the severity of tricuspid regurgitation by comparing the SV calculated with SV_RV_ and the true SV (i.e., through the pulmonary artery) estimated with another method such as SV_(RPA+LPA)_.

As shown in Table 5 and Figure 4, SV_(RPA+LPA)_ was the only surrogate of COp that reached the two prespecified interchangeability criteria with two SV surrogates of COs (SV_AAO_ and SV_(DOA+SVC)_). The agreement between SV_(RPA+LPA)_ and SV_(DOA+SVC)_ even reached the more restrictive agreement necessary for the acceptance of a new method with a gold standard method (2SD of bias for SV ≤ 14.3 and PE ≤ 20%). In this context, SV_(RPA+LPA)_ may be the most promising method for SV determination in patients with PH. In comparison to SV_PA_, it might be less influenced by (1) vortexes, which mainly appear in the main PA; (2) main PA dilatation with possible issues in contour determination; and (3) pulmonary regurgitation. Indeed, the main PA acts as a blood reservoir during the diastolic time, and the pulmonary regurgitation would probably be of lesser impact if the measurement is made distally to the main PA. Even though this seems the most promising method for SV measurement, this needs to be prospectively validated against the gold standard DF and in patients with significant haemodynamic PH severity.

The main limitations of our study include the small number of subjects, the mildly elevated mean mPAP, the retrospective analysis, the delay between MRI and RHC, and the absence of a direct comparison to a gold standard DF. However, TD is widely used, and thus our data may be considered a comparison with a real-life setting. The strengths of our study are related to the rigorous methodology with a predetermined cut-off for the 2SD of bias, which is unfortunately rarely used in this field, and the use of new localisations for flow determination in PH, with the measurement in RPA + LPA for the pulmonary circulation and DAO + SVC for the systemic circulation.

## 5. Conclusions

We could not demonstrate the interchangeability of SV_MRI_ and SV_TD_, but this is probably due to an overestimation of TD precision in PH. Estimation of COp with SV_(RPA+LPA)_ was shown to agree more closely to methods of COs estimation than the previously described methods for COp determination in this population (SV_RV_ and SV_PA_). SV_(RPA+LPA)_ may be the best non-invasive MRI method to determine COp in precapillary PH.

## Figures and Tables

**Figure 1 jcm-11-02717-f001:**
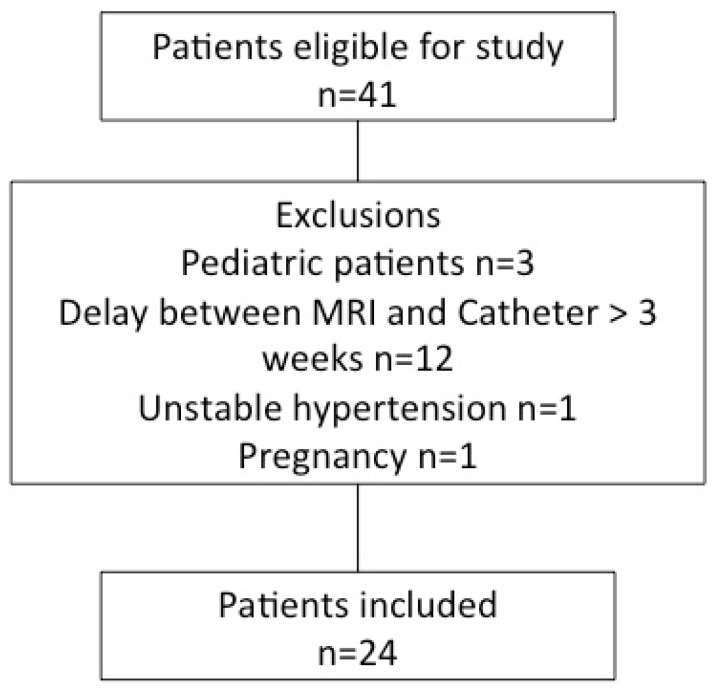
Flow chart.

**Figure 2 jcm-11-02717-f002:**
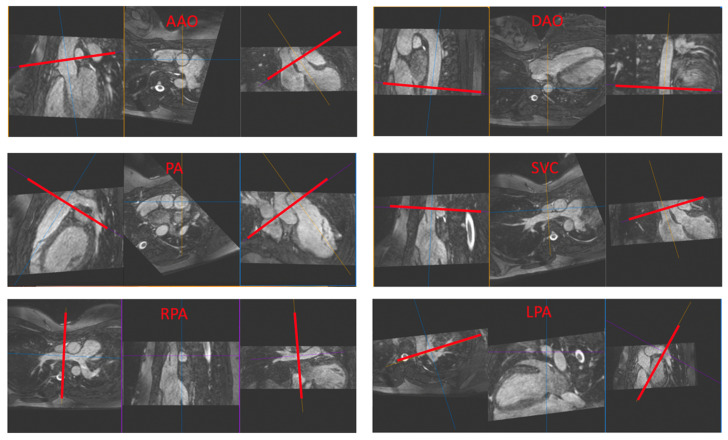
Placement of images for the major vessels, the resulting phase-contrast flow image, and the calculated flow curve. The green circle is placed in the AAO in this figure. Flow quantification using Syngovia was semi-automated. AAO: ascending aorta, DAO: descending aorta, LPA: left pulmonary artery, PA: pulmonary artery, RPA: right pulmonary artery, SVC: superior vena cava.

**Figure 3 jcm-11-02717-f003:**
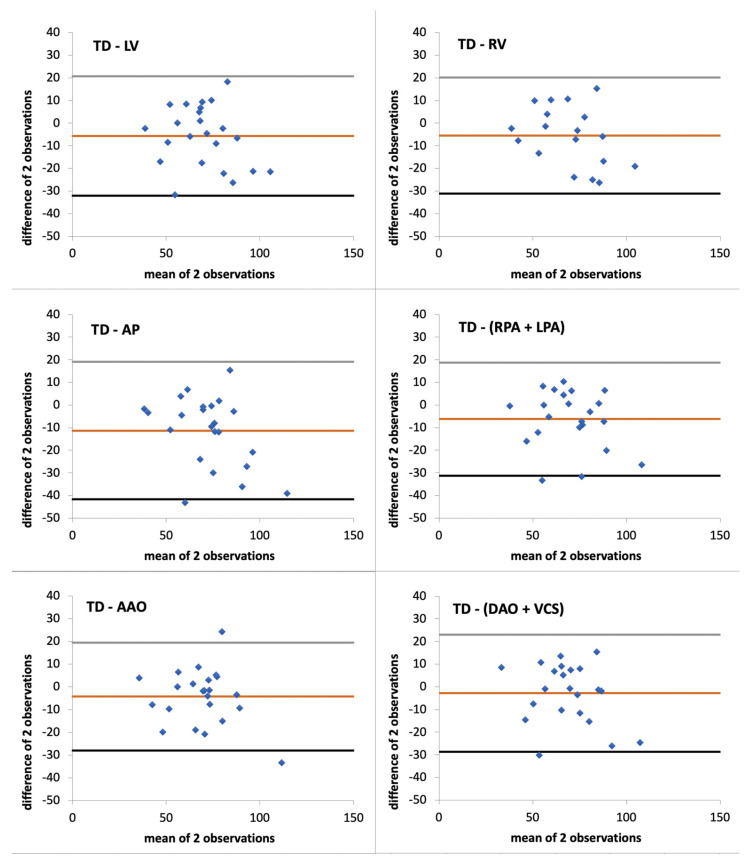
Bland and Altman analyses of SV_MRI_ against SV_TD_. LoA are in grey and black. Bias is in orange. Results are in stroke volume (mL/beat). AAO: ascending aorta, DAO: descending aorta, LV: left ventricle, LPA: left pulmonary artery, PA: pulmonary artery, RV: right ventricle, RPA: right pulmonary artery, TD: thermodilution, SVC: superior vena cava.

**Figure 4 jcm-11-02717-f004:**
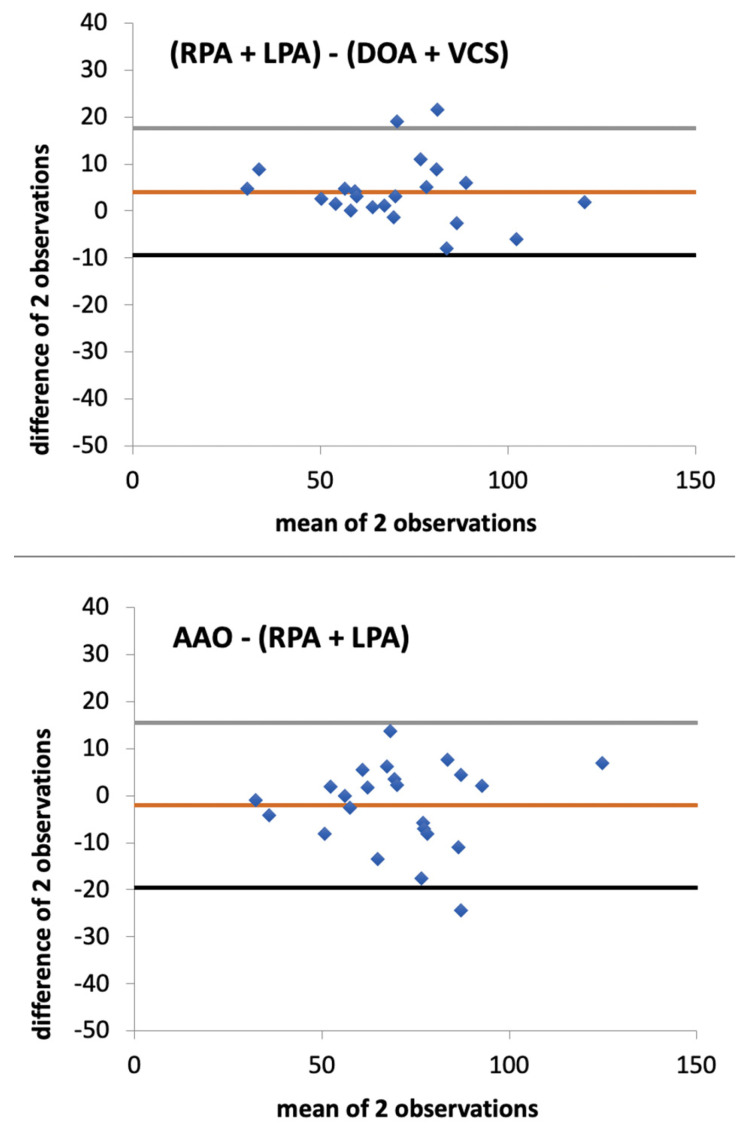
Bland and Altman analyses of SV_(RPA+LPA)_ against SV_(DOA+VCS)_ and SV_AAO_. LoA are in grey and black. Bias is in orange. Results are in stroke volume (mL/beat). AAO: ascending aorta, DAO: descending aorta, LPA: left pulmonary artery, RPA: right pulmonary artery, SVC: superior vena cava.

**Table 1 jcm-11-02717-t001:** Patient characteristics.

Parameter	Value
Total patients	*n* = 24
Male:Female	9:15
Age at MRI	60 ± 14
Interval between MRI and RHC	5.8 ± 5.6
mPAP (mmHg)	29 ± 15
mPAP < 25 mmHg (no PH)	*n* = 12
mPAP ≥ 25 mmHg (with PH)	*n* = 12
CI (TD) (L min^−1^ m^−2^)	2.5 ± 0.7
PVR (WU)	5.3 ± 4.1
TAPSE (mm)	19 ± 4
HR (MRI) (beats/min)	73 ± 13
HR (RHC) (beats/min)	70 ± 9
PH group 1	*n* = 4
PH group 4	*n* = 7
Other	*n* = 1

Data are in mean ± SD unless otherwise stated. CI: cardiac index; HR: heart rate; mPAP: mean pulmonary arterial pressure; PH: pulmonary hypertension; PVR: pulmonary vascular resistance; TAPSE: tricuspid annular plan systolic excursion.

**Table 2 jcm-11-02717-t002:** Inter- and intra-observer ICC.

ICC	AAO	DAO	SVC	PA	LPA	RPA
Inter	0.981	0.983	0.826	0.926	0.889	0.901
Intra	0.995	0.991	0.995	0.998	0.996	0.999

AAO: ascending aorta, DAO: descending aorta, ICC: intraclass correlation; inter: inter-observer; intra: intra-observer LPA: left pulmonary artery, PA: pulmonary artery, RPA: right pulmonary artery, SVC: superior vena cava.

**Table 3 jcm-11-02717-t003:** Mean and standard deviation values for CO and SV.

	CO L/min	SV mL/Beat
TD	4.7 ± 1.0	67 ± 16
LV	5.1 ± 1.1	71 ± 19
RV	5.1 ± 1.2	73 ± 21
AAO	4.9 ± 1.2	70 ± 20
PA	5.5 ± 1.4	76 ± 23
RPA + LPA	5.1 ± 1.1	71 ± 20
DAO + SVC	4.8 ± 1.2	69 ± 21

Data are in mean ± SD. AAO: ascending aorta, CO: cardiac output, DAO: descending aorta, LV: left ventricle, LPA: left pulmonary artery, PA: pulmonary artery, RV: right ventricle, TD: thermodilution; RPA: right pulmonary artery, SV: stroke volume, SVC: superior vena cava.

**Table 4 jcm-11-02717-t004:** Comparison between the six different SV_MRI_ methods against SV_TD_.

	Bland and Altman Analysis			Linear Regression with Coefficient of Correlation (r), Slope (a), and Intercept (b) with SV_MRI_ on the y-Axis and Axis SV_TD_ on the x-Axis		
Compared SV_MRI_ Method	Bias, mL/Beat	2SD of Bias, mL/Beat	PE (%)	r	a	b mL/Beat
LV	−5.6	±26.9	38.5	0.80	0.93	11.8
RV	−5.5	±26.2	37.5	0.87	0.91	9.7
AAO	−4.3	±24.1	34.9	0.65	0.91	17.3
PA	−11.3	±31.1	42.8	0.65	0.82	16.9
RPA + LPA	−6.2	±25.4	36.3	0.66	0.33	18.3
DAO + SVC	−2.9	±26.3	38.2	0.61	0.85	14.3

AAO: ascending aorta, CO: cardiac output, DAO: descending aorta, LV: left ventricle, LPA: left pulmonary artery, MRI: magnetic resonance imaging, PA: pulmonary artery, PE: percentage error, RV: right ventricle, RPA: right pulmonary artery, SD: standard deviation, SV: stroke volume, SVC: superior vena cava, thermodilution: TD.

**Table 5 jcm-11-02717-t005:** Fifteen paired comparisons of the six different MRI methods.

	Bland and Altman Analysis			Linear Regression with Coefficient of Correlation (r), Slope (a), and Intercept (b) with the First Method on the y-Axis and the Second Method on the x-Axis		
Compared SV_MRI_ (First Method/Second Method)	Bias, mL/Beat	2SD of Bias, mL/Beat	PE (%)	r	a	b, mL/Beat
RV/PA	−6.4	±23.7	31.3	0.89	0.83	7.8
RV/(RPA + LPA)	−3.1	±18.3	**25.2**	0.91	0.91	4.9
RV/AAO	0.5	±21.6	**29.9**	0.88	0.91	7.7
LV/PA	−5.1	±29.1	39.4	0.85	0.78	12.8
LV/(RPA + LPA)	−0.9	±24.8	34.9	0.88	0.86	10.1
LV/AAO	1.7	±21.8	31.0	0.90	0.88	10.9
AAO/(DAO + SVC)	1.7	±19.4	**27.9**	0.94	0.86	10.3
PA/(DAO + VCS)	8.6	±18.3	**25.1**	0.93	0.91	13.6
**(RPA + LPA)/(DAO + VCS)**	4.1	±**13.8**	**19.7**	0.97	0.89	11.0
PA/(RPA + LPA)	5.0	± 19.8	**26.8**	0.93	0.99	4.8
PA/AAO	−6.8	±18.7	**25.6**	0.93	0.99	6.9
**(RPA + LPA) /AAO**	−2.1	±**17.9**	**25.5**	0.94	0.94	6.3
**LV/RV**	−0.5	±**17.1**	**23.7**	0.91	0.99	1.5
LV/(DAO + VCS)	3.5	±23.8	33.9	0.83	0.89	4.3
RV/(DAO + VCS)	2.0	±18.4	**25.1**	0.90	0.92	3.6

Bold values meet the predetermined interchangeability criteria either for PE or for 2 SD of bias. Methods in bold are meeting both interchangeability criteria. AAO: ascending aorta, DAO: descending aorta, LV: left ventricle, LPA: left pulmonary artery, MRI: magnetic resonance imaging, PA: pulmonary artery, PE: percentage error, RV: right ventricle, RPA: right pulmonary artery, SD: standard deviation, stroke volume: SV, SVC: superior vena cava.

## Data Availability

Data can be shared upon reasonable request.

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
