# Peer review of "Non-Invasive Cardiac Output Determination Using Magnetic Resonance Imaging and Thermodilution in Pulmonary Hypertension"

_jcm, 2022, doi:10.3390/jcm11102717_

Round 1

Reviewer 1 Report

Paper is important for cardiology. Following revisions are to be incorporated which is expectation of the reviewer.
(1) What were the parameters which were selected for improving the performance?
(2) How simulation was accomplished?
(3) What were the major challenges in the proposed research work?
(4) Is the proposed technique useful for wide spectrum of patient s?
(5) Author should add the motivations, problem, and solution statement in the abstract.
(6) All tables and figures should be explained clearly.
(7) The English and typo errors of the paper should be checked in the presence of native English speaker.
(8) All equations should be clearly explained with explanation on all associated variables.
(9) Author should add one Section "Related Work" in the paper.
(10) The methodology of the paper should be clearly explained with appropriate flow charts.
(11) Highlight the more applications of the proposed technique.
(12) What are motivations behind this research work?
(13) Add more explanation on obtained results with critical analysis.
(14) Literature survey is not upto the mark. Author must cite suggested papers for enhancing the quality of the
paper-
(a) Identification of Human Vital Functions Directly Relevant to the Respiratory System Based on the Cardiac and Acoustic Parameters and Random Forest
(b) Electrocardiogram Data Compression Techniques for Cardiac Healthcare Systems: A Methodological Review
(c) Segmentation Integrating Watershed and Shape Priors Applied to Cardiac Delayed Enhancement MR Images
(d) Neuromonitoring par la spectroscopie dans le proche infrarouge en chirurgie cardiaque pédiatrique:
Neuromonitoring by near infrared spectroscopy in paediatric cardiac surgery
(e) ABYSS: Therapeutic hypothermia by total liquid ventilation following cardiac arrest and resuscitation
(f) Three-Dimensional Computational Modeling of an Extra-Descending Aortic Assist Device Using Fluid-
Structure Interaction
(g) A novel method of cardiac arrhythmia detection in electrocardiogram signal
(h) A 3D Network Based Shape Prior for Automatic Myocardial Disease Segmentation in Delayed-
Enhancement MRI
(i) A framework for multimodal imaging-based prognostic model building: Preliminary study on multimodal
MRI in Glioblastoma Multiforme
(j) ROI-Based Compression Strategy of 3D MRI Brain Datasets for Wireless Communication

Author Response

Dear Editor,

We would like to thank the reviewers for the rapid and constructive review of our work.

For best clarity, we would like to answer the reviewer’s questions as below:

Reviewer 1:

(1) What were the parameters which were selected for improving the performance?

In this study, we decided to favour the RPA (Right Pulmonary Artery) + LPA (Left Pulmonary Artery). These parameters demonstrated the best performance. Please refer to the manuscript, section comparison in-between different MRI methods in the discussion.

(2) How simulation was accomplished?

There was no simulation in our study. The flow values were determined post acquisition as described in the materials and methods in the section CO determined by MRI.

(3) What were the major challenges in the proposed research work?

The major challenge was the expertise needed to perform MRI data analysis in this specific population.

(4) Is the proposed technique useful for wide spectrum of patients?

Non-invasive cardiac output determination would be of high value for patients with PH especially for group 1 and group 4 patients for prognosis determination and for diagnosis assessment (especially now that PVR has been suggested as an obligatory criterion for precapillary PH definition). CO determined in RPA and LPA would be especially useful for patients with shunt in order to determine the magnitude of the shunt (Qs/Qp ratio) and to determine the correct PVR for therapeutic purpose. This is described in our discussion in the section 4.3 (about the comparison between different MRI methods).

(5) Author should add the motivations, problem, and solution statement in the abstract.

The abstract is restricted to 200 words and we felt that it would be more appropriate to dedicate this part to the most relevant aspects of our work.

(6) All tables and figures should be explained clearly.

We thank the reviewer for the remark. We have adapted the legend of the figures and tables accordingly.

(7) The English and typo errors of the paper should be checked in the presence of native English speaker.

We thank the reviewer to consider that english language and style needed minor spell check. The first author of the paper is native English speaker.

(8) All equations should be clearly explained with explanation on all associated variables.

We controlled that all equations were appropriately explained (materials and methods sections).

(9) Author should add one Section "Related Work" in the paper.

We thank the reviewer for this proposal. We have thoroughly reviewed the literature of CO determination using MRI (cf reference 18, 19, 25, 26, 27). We feel that this related work review developed in the introduction and the discussion offer a comprehensive overview of the field.

(10) The methodology of the paper should be clearly explained with appropriate flow charts.

We thank the reviewer for this remark. However we feel that the methodology of our study is clear and flow chart use appropriate.

(11) Highlight the more applications of the proposed technique.

We are unsure to understand the remark. We have detailed in point (4) the use of CO in PH.

(12) What are motivations behind this research work?

CO determination is mandatory for PH patients and a non-invasive method would diminish the need of repetitive invasive assessment. 

(13) Add more explanation on obtained results with critical analysis.

This proposal is nonspecific and we think that our results are critically analysed.

(14) Literature survey is not up to the mark. Author must cite suggested papers for enhancing the quality of thepaper

We are in disagreement with the reviewer. Indeed, none of the proposed reference is linked to our work.

Reviewer 2 Report

Dear author

This article aims to clarify that MRI can be used to measure cardiac output non-invasively, which would be very interesting. However, because of overestimated TD, the aims of this study is not enough to be attained, and your results and conclusions would have low impact.

1) TD should be re-measured and re-analyzed with the findings of MRI.

Sincerely,

From reviewer.

Author Response

Dear Editor,

We would like to thank the reviewers for the rapid and constructive review of our work.

For best clarity, we would like to answer the reviewer’s questions as below:

Reviewer 2 :

  • TD should be re-measured and re-analyzed with the findings of MRI

We thank the reviewer for the careful review. However we do not agree with his proposal. TD was performed according to strict standard methodology as proposed in the current international guidelines for the diagnosis and treatment of PH (Galiè et al 2015). Furthermore, right heart catheterisation is performed by experienced clinician. We do not think that TD is overestimated. However data are accumulating concerning TD precision in PH patients showing that TD precision is probably less precise than initially demonstrated (Hoeper et al 1999). The time delay between the present MRI analysis and new TD measurement would be unacceptable for a proper analysis and comparison.

Sincerly, 

Léon Genecand on the behalf of the co-authors.

Round 2

Reviewer 1 Report

Accepted in current form.

Reviewer 2 Report

Dear author

Thank you very much for your detailed comments.

I accept the author comments because the author said that TD was performed according to the current international guidelines of PH, and right heart catheterisation is performed by experienced clinician. I think that the validity of TD would be ensured.

Sincerely,